# Bridging Continuous and Discrete Spaces: Interpretable Sentence Representation Learning via Compositional Operations

**James Y. Huang**[†], **Wenlin Yao**[‡], **Kaiqiang Song**[‡], **Hongming Zhang**[‡],
**Muhao Chen**[†] and **Dong Yu**[‡]

[†]University of Southern California;  [‡]Tencent AI Lab, Seattle

{huangjam,muhaoche}@usc.edu;
{wenlinyao, riversong, hongmzhang, dyu}@global.tencent.com

## Abstract

Traditional sentence embedding models encode sentences into vector representations to capture useful properties such as the semantic similarity between sentences. However, in addition to similarity, sentence semantics can also be interpreted via compositional operations such as sentence fusion or difference. It is unclear whether the compositional semantics of sentences can be directly reflected as compositional operations in the embedding space. To more effectively bridge the continuous embedding and discrete text spaces, we explore the plausibility of incorporating various compositional properties into the sentence embedding space that allows us to interpret embedding transformations as compositional sentence operations. We propose INTERSENT, an end-to-end framework for learning interpretable sentence embeddings that supports compositional sentence operations in the embedding space. Our method optimizes operator networks and a bottleneck encoder-decoder model to produce meaningful and interpretable sentence embeddings. Experimental results demonstrate that our method significantly improves the interpretability of sentence embeddings on four textual generation tasks over existing approaches while maintaining strong performance on traditional semantic similarity tasks.[1].

## 1 Introduction

Learning universal sentence embeddings is crucial to a wide range of NLP problems, as they can provide an out-of-the-box solution for various important tasks, such as semantic retrieval (Gillick et al., 2018), clustering (Hadifar et al., 2019), and question answering (Nakov et al., 2016). Recently, contrastive learning has been shown to be an effective training paradigm for learning sentence embeddings (Giorgi et al., 2021; Yan et al., 2021; Gao et al., 2021; Chuang et al., 2022). These methods optimize the sentence representation space such that the distance between embeddings reflects the semantic similarity of sentences.

While the similarity structure of sentence embedding models is an important aspect of the inter-sentence relationship, contrastive learning methods do not provide a direct way of interpreting the information encoded in the sentence embedding. Despite the existence of probes for individual linguistic properties (Conneau et al., 2018), it is still unclear whether the embedding fully captures the semantics of the original sentence necessary for reconstruction. Moreover, sentence semantics not only can be interpreted by their similarity but also via sentence operations such as fusion, difference and compression. While these operations of sentence semantics have been previously studied individually as sequence-to-sequence generation tasks (Geva et al., 2019; Botha et al., 2018; Filippova and Altun, 2013; Rush et al., 2015), it remains an open research question whether these operations can be directly captured as operations in the sentence embedding space. We argue that the ability to interpret and manipulate the encoded semantics is an important aspect of interpretable sentence embeddings, which can bridge the continuous embedding space and the discrete text space. Particularly, this ability benefits concrete tasks such as multi-hop search and reasoning (Khattab et al., 2021), instruction following (Andreas and Klein, 2015), compositional generation (Qiu et al., 2022), and summarization (Brook Weiss et al., 2022).

In this work, we propose INTERSENT, an end-to-end framework for learning interpretable and effective sentence embeddings that supports compositional sentence operations. Our method combines both generative and contrastive objectives to learn a well-structured embedding space that satisfies useful properties for both utility and interpretability. Specifically, together with an encoder-decoder model, we train several small operator networks on

---

[1]Code is available at https://github.com/jyhuang36/InterSent

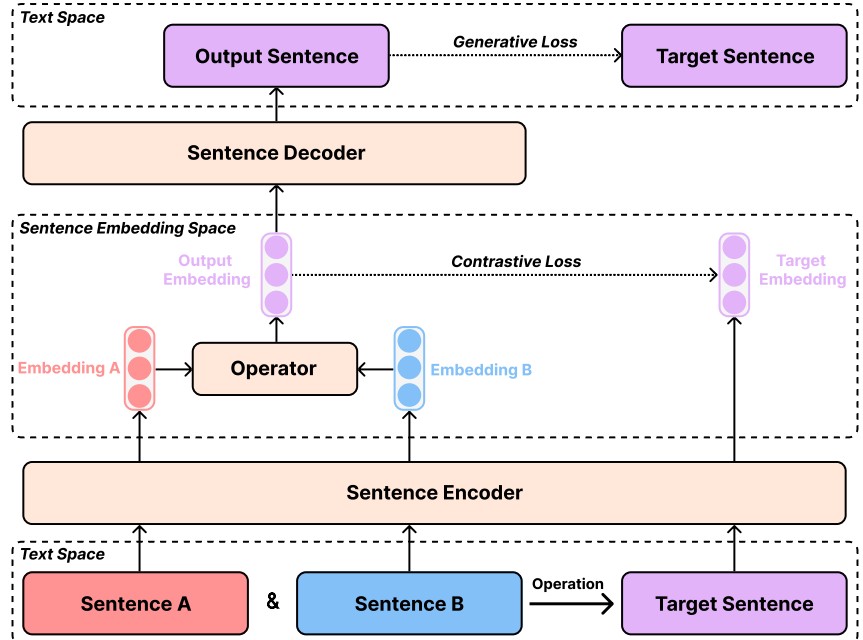

Figure 1: Overview of our proposed framework INTERSENT for learning interpretable sentence embeddings. Given a pair of input sentences (or a single input sentence for compression and reconstruction), INTERSENT encodes the input sentence(s) into sentence embedding (s), which are fed into an operator network to compute the output embedding for decoding. During training, INTERSENT ensures alignment between the output and target in both the embedding space and text space via contrastive and generative objectives respectively.

easily available weak supervision data to capture different sentence operations. Sentence embeddings learned by our model not only preserve the ability to express semantic similarity but also support various sentence operations (such as sentence fusion, difference, and compression) for interpreting compositional semantics.

Our contributions are three-fold. First, we propose INTERSENT, an interpretable sentence embedding model that establishes a mapping between the continuous embedding space and the discrete text space by connecting transformations on embeddings and compositional operations on texts. Second, our method significantly improves the interpretability of sentence embeddings on four textual generation tasks. Third, we demonstrate that interpretable sentence embeddings learned by our method still maintain strong performance on traditional semantic similarity and text retrieval tasks.

## 2 Method

Fig. 1 illustrates the overall architecture of INTERSENT. Our method, INTERSENT, optimizes both a contrastive objective (in the continuous space) and a generative objective (in the discrete text space) jointly during training. In this section, we define the notion of interpretability for a sen-

tence embedding space and provide a detailed description of our proposed framework INTERSENT.

### 2.1 Problem Definition

Our notion of interpretable sentence representations centers around the ability to interpret embedding vectors and simple transformations defined over them in the embedding space, as human-comprehensible sentences and sentence operations in the text space. In other words, our goal is to establish

- a mapping between embedding vectors and sentences, which allows us to both encode sentences into vectors and decode vectors into sentences;

- a mapping between certain simple transformations over vectors and certain sentence operations, which allows us to manipulate sentence semantics in the embedding space.

In this work, we explore the plausibility of supporting several common sentence operations that are previously studied as individual sequence-to-sequence generation tasks (Geva et al., 2019; Botha et al., 2018; Filippova and Altun, 2013; Rush et al., 2015). These include the following compositional operations:

- *Sentence Fusion*: Given the embeddings of two sentences, an embedding of their fusion, which contains information from both sentences, can be inferred.
- *Sentence Difference*: The embedding of the difference between two sentences can be inferred from their individual embeddings. To avoid ambiguity, we restrict this definition to only cases where the first sentence contains the whole information of the second sentence. In other words, sentence difference is essentially defined as an inverse operation of sentence fusion.

In addition, we consider the following compression operation, as well as sentence reconstruction that help interpret the meaning of any sentence embedding vector.

- *Sentence Compression*: Given the embedding of a sentence, we seek to infer the embedding of the compression or summarization of this sentence.
- *Sentence Reconstruction*: The content of the original sentence can be recovered from its sentence embedding. This property serves as the foundation for interpretability, as it allows us to understand the semantics of any sentence embedding vector, including those computed by applying the sentence operations (in the form of vector transformations) in the embedding space.

In other words, we aim to learn a sentence encoder $Enc$, a sentence decoder $Dec$, and sentence operator functions $f_{\text{fus}}$, $f_{\text{diff}}$, $f_{\text{comp}}$ that satisfies the following properties:

- $Enc(s_1 \oplus s_2) \approx f_{\text{fus}}(Enc(s_1), Enc(s_2))$ where sentence $s_1 \oplus s_2$ is the fusion of sentence $s_1$ and $s_2$.
- $Enc(s_1 \ominus s_2) \approx f_{\text{diff}}(Enc(s_1), Enc(s_2))$ where sentence $s_1 \ominus s_2$ is the difference of sentence $s_1$ and $s_2$.
- $Enc(\bar{s}) \approx f_{\text{comp}}(Enc(s))$ where sentence $\bar{s}$ is a compression of sentence $s$.
- $s' \approx Dec(Enc(s))$ where $s'$ and $s$ are a pair of sentences expressing the same semantics.

## 2.2 Sentence Operator

We use two-layer MLPs to fit operator functions over embeddings, which can be trained together with the rest of the model components in an end-to-end manner. Some compositional sentence operations may be alternatively approximated with simple arithmetic operations, such as addition and subtraction of embedding vectors. We empirically show that defining sentence operations with simple arithmetics leads to inferior performance on downstream tasks (see §4.2 for more details). In comparison, MLP transformations achieve a good balance between simplicity and flexibility to fit different sentence operations. All operator networks take a sentence embedding (for compression), or a concatenated pair of sentence embeddings (for fusion and difference), and compute a new sentence embedding as the target. For the compression operator, we use a small intermediate dimension size to limit the information flow and encourage the model only to preserve essential information in the compressed embeddings.

## 2.3 Bottleneck Model

Our model uses Transformer-based language models as encoders and decoders for sentence embeddings. Unlike in the typical encoder-decoder architecture for sequence generation, where the decoder has access to the contextualized representations of all tokens, the encoder in INTERSENT only outputs a single vector as the representation for each input sentence. Following previous work (Gao et al., 2021; Chuang et al., 2022), we take the representation of the [CLS] token from the encoder as the sentence embedding. This information bottleneck forces the model to produce meaningful sentence embeddings such that the decoder can reconstruct the semantics given the embedding vectors alone. It is worth noting that the encoder and decoder are shared across all sentence operations, which forces the model to capture the operations in the embedding space, rather than learning task-specific encoders and decoders for each operation.

## 2.4 Training Objective

INTERSENT combines contrastive and generative objectives to learn interpretable sentence embeddings that captures both semantic similarity and sentence operations. Specifically, we train INTERSENT to maximize alignment between outputs and targets in both the embedding space and text space. This means that the output embedding computed by an operator function should be close to the target embedding, and the target sentence can be decoded from the output embedding. The first objective is realized by optimizing a contrastive loss with in-batch negatives. For the $i$-th training instance, let $\mathbf{v}_i$ and $\mathbf{v}_i^+$ denote the output embed-

ding (computed by the encoder $f_{enc}$ and an operator function from the input sentence(s)) and the target embedding (computed by encoding the target sentence directly). The contrastive objective for $(\mathbf{v}_i, \mathbf{v}_i^+)$ is given by

$$\mathcal{L}_{i,con} = -\log \frac{e^{\text{sim}(\mathbf{v}_i, \mathbf{v}_i^+)/\tau}}{\sum_{j=1}^{N} e^{\text{sim}(\mathbf{v}_i, \mathbf{v}_j^+)/\tau}}, \quad (1)$$

where $N$ is the mini-batch size, and $\tau$ is the softmax temperature. To ensure the target sentence can be decoded from the output embedding, we also optimize the following conditional generative loss:

$$\mathcal{L}_{i,gen} = -\frac{1}{|T_i|} \sum_{k=1}^{|T_i|} \log p(t_{i,k}|T_{i,<k}, \mathbf{v}_i). \quad (2)$$

where $T_i$ denotes the target sequence for the $i$-th training instance in the batch. Both losses are combined with a balancing factor $\alpha$:

$$\mathcal{L}_i = \mathcal{L}_{i,con} + \alpha \mathcal{L}_{i,gen}. \quad (3)$$

# 3 Experiments

In this section, we evaluate the interpretability of INTERSENT and recent sentence embedding models on four generative sentence operation tasks. Then, we conduct experiments on traditional zero-shot semantic textual similarity benchmarks. Finally, we compare our model with previous methods on zero-shot sentence retrieval tasks.

## 3.1 Data

To learn the sentence operations we described previously, INTERSENT is trained on the combination of five weak supervision datasets for sentence fusion, difference, compression, and reconstruction. **DiscoFuse** (Geva et al., 2019) is a sentence fusion dataset constructed with heuristic rules that combines two sentences into one. **WikiSplit** (Botha et al., 2018) is a split-and-rephrase dataset extracted from Wikipedia edits, which we use for the sentence difference task. For compression, we use the **Google** (Filippova and Altun, 2013) and **Gigaword** (Napoles et al., 2012) sentence compression datasets. Both of these datasets are collected by pairing the news headline with the first sentence of the article. Finally, we use **ParaNMT** (Wieting and Gimpel, 2018) for sentence reconstruction, which contains paraphrase pairs generated from back-translation. It is worth noting that all of these datasets are constructed automatically. More details of the datasets can be found in Appx. §A.

## 3.2 Implementation

We train the encoder and decoder with weights initialized from RoBERTa[2] (Liu et al., 2019) and BART (Lewis et al., 2020), respectively. This hybrid setting allows us to utilize the high-quality sentence-level representations from RoBERTa while taking advantage of the generative capability of BART. In experiments, we also adopt another two encoder-decoder model backbones (i.e., T5 (Raffel et al., 2020) and BERT+BERT), but they perform slightly worse than RoBERTa+BART (see Appx. §C for more details). The loss balancing factor is set to 0.01, as the contrastive loss converges much faster than the generative loss. We train the model on the combination of the five datasets for five epochs. More details about hyperparameters can be found in Appx. §B.

## 3.3 Baseline

We compare our model with previous sentence embedding models, as well as unsupervised and contrastive baselines trained on the same datasets as INTERSENT. All models, including INTERSENT, use RoBERTa-base as the sentence encoder. **RoBERTa-cls** and **RoBERTa-avg** are sentence embeddings directly extracted via the [CLS] token and average pooling from an unfinetuned RoBERTa model. **SRoBERTa** (Reimers and Gurevych, 2019) trains a sentence embedding model on supervised NLI pairs by optimizing a cross-entropy loss to predict NLI labels. **De-CLUTR** (Giorgi et al., 2021) is an unsupervised contrastive sentence embedding model trained on sampled positive and negative pairs from raw text. **SimCSE** (Gao et al., 2021) instead uses different dropout masks applied to the same sentence as data augmentation. **DiffCSE** (Chuang et al., 2022) introduces an additional replaced token detection objective such that the embeddings are sensitive to token replacement. Since previous methods are trained on different datasets, we additionally include unsupervised and supervised contrastive models trained on our data for direct comparison. The **Unsupervised Contrastive** baseline is trained similarly to SimCSE, by breaking down all sentence pairs and triplets in our datasets into individual sentences. For (weakly) **Supervised Contrastive** baseline, we concatenate the two shorter sentences in the fusion and difference datasets and then use all sentence

---

[2]We follow previous work and use RoBERTa-base model (110M parameters) to make the results comparable.

| Model | Fusion | | | Difference | | | Comp. (Google) | | | Comp. (Gigaword) | | | Avg. | | |
|---|---|---|---|---|---|---|---|---|---|---|---|---|---|---|---|
| | R-1 | R-2 | R-L | R-1 | R-2 | R-L | R-1 | R-2 | R-L | R-1 | R-2 | R-L | R-1 | R-2 | R-L |
| RoBERTa-cls | 42.8 | 16.9 | 35.1 | 30.9 | 10.4 | 27.0 | 27.7 | 10.6 | 25.8 | 30.4 | 11.5 | 27.7 | 32.9 | 12.4 | 28.9 |
| RoBERTa-avg | 68.7 | 43.1 | 58.5 | 54.4 | 29.0 | 46.3 | 43.6 | 23.8 | 41.2 | 37.4 | 16.4 | 34.1 | 51.0 | 28.1 | 45.0 |
| SRoBERTa | 49.0 | 21.9 | 39.0 | 39.1 | 16.2 | 33.2 | 37.8 | 19.1 | 35.5 | 36.0 | 15.5 | 32.7 | 40.5 | 18.2 | 35.1 |
| DeCLUTR | 73.4 | 46.7 | 61.6 | 56.1 | 30.1 | 47.3 | 50.3 | 30.1 | 47.9 | 39.6 | 17.9 | 36.1 | 54.9 | 31.2 | 48.2 |
| SimCSE | 53.2 | 24.3 | 42.0 | 36.1 | 13.6 | 30.8 | 38.4 | 18.2 | 35.9 | 35.0 | 14.2 | 31.5 | 40.7 | 17.6 | 35.1 |
| DiffCSE | 57.8 | 28.7 | 46.0 | 40.3 | 16.9 | 34.2 | 41.8 | 21.0 | 39.0 | 36.4 | 15.3 | 32.8 | 44.1 | 20.5 | 38.0 |
| *Encoders trained on our data* | | | | | | | | | | | | | | | |
| Unsup. Contr. | 57.9 | 28.6 | 45.8 | 38.9 | 15.7 | 33.1 | 41.4 | 20.7 | 38.7 | 35.9 | 15.0 | 32.4 | 43.5 | 20.0 | 37.5 |
| Sup. Contr. | 57.7 | 29.4 | 46.9 | 36.9 | 14.4 | 31.7 | 50.3 | 28.7 | 47.5 | 41.6 | 19.2 | 37.9 | 46.6 | 22.9 | 41.0 |
| InterSent | **88.7** | **71.9** | **82.2** | **73.0** | **48.4** | **64.3** | **69.6** | **50.9** | **66.3** | **48.0** | **24.7** | **43.8** | **69.8** | **51.5** | **64.2** |

Table 1: Model performance on four textual generation tasks for interpretability evaluation. Unsup. Contr. and Sup. Contr. represents Unsupervised and Supervised Contrastive baselines respectively. We report ROUGE-1/2/L scores.

| Model | STS12 | STS13 | STS14 | STS15 | STS16 | STS-B | SICK-R | Avg. |
|---|---|---|---|---|---|---|---|---|
| RoBERTa-cls | 16.67 | 45.57 | 30.36 | 55.08 | 56.99 | 38.82 | 61.90 | 43.63 |
| RoBERTa-avg | 32.11 | 56.33 | 45.22 | 61.35 | 61.98 | 55.49 | 62.03 | 53.49 |
| SRoBERTa[†] | 71.54 | 72.49 | 70.80 | 78.74 | 73.69 | 77.77 | 74.46 | 74.21 |
| DeCLUTR[‡] | 52.41 | 75.19 | 65.52 | 77.12 | 78.63 | 72.41 | 68.62 | 69.99 |
| SimCSE[‡] | 70.16 | 81.77 | 73.24 | 81.36 | 80.65 | 80.22 | 68.56 | 76.57 |
| DiffCSE[◇] | 70.05 | **83.43** | 75.49 | 82.81 | **82.12** | 82.38 | 71.19 | 78.21 |
| *Encoders trained on our data* | | | | | | | | |
| Unsupervised Contrastive | 69.11 | 82.05 | 74.99 | 82.65 | 81.00 | 81.20 | 69.56 | 77.22 |
| Supervised Contrastive | **72.46** | 81.42 | **77.18** | **84.08** | 79.68 | **82.95** | **74.82** | **78.94** |
| InterSent | 70.97 | 81.03 | 75.30 | 83.18 | 79.60 | 80.60 | 69.45 | 77.16 |

Table 2: Model performance on Semantic Textual Similarity (STS) tasks. We report Spearman's correlation on all tasks. †: results taken from (Reimers and Gurevych, 2019). ‡: results taken from (Gao et al., 2021). ◇: results taken from (Chuang et al., 2022).

| Model | MRR@10 | recall@10 |
|---|---|---|
| RoBERTa-cls | 56.53 | 65.43 |
| RoBERTa-avg | 54.87 | 64.89 |
| SRoBERTa | 71.13 | 79.81 |
| DeCLUTR | 75.84 | 87.02 |
| SimCSE | 79.78 | 89.32 |
| DiffCSE | 79.49 | 89.53 |
| *Encoders trained on our data* | | |
| Unsupervised Contrastive | 79.34 | 89.11 |
| Supervised Contrastive | 79.67 | 89.71 |
| InterSent | **80.30** | **89.94** |

Table 3: Model performance on the zero-shot QQP sentence retrieval task. We report both Mean Reciprocal Rank@10 (MRR@10) and recall@10.

pairs as weak supervision pairs.

## 3.4 Interpretability

**Setup.** We first compare the interpretability of sentence embedding space on generative sentence operation tasks including fusion, difference and compression. Since none of the baseline models include a decoder for sentence generation, we stack operator networks and decoders on top of their trained encoders, making the model architecture identical across all models. For all baseline models, we take the sentence embeddings encoded by these sentence encoders, and optimize the added operator networks and decoder during training. This setting allows us to examine if existing sentence embeddings already support sentence operations and contain sufficient information for reconstruction. By comparing contrastive baselines with our method trained on the same data, we can also have a better understanding of how much fine-tuning sentence encoders (along with the rest of the model) on both generative and contrastive objectives can benefit the interpretability of the learned embedding space. We report ROUGE-1/2/L scores (Lin, 2004).

**Results.** As shown in Tab. 1, our method significantly outperforms all baselines across four sentence operation tasks. Without fine-tuning, average pooling, which aggregates all token representations, unsurprisingly outperforms CLS pooling by a large margin. Among previous sentence embedding mod-

els, DeCLUTR, which incorporates a masked language modeling objective, has the best overall interpretability performance. While contrastive learning on our data indeed helps the model adapt to our datasets, there still exists a large gap between the supervised contrastive baseline and our INTERSENT. This demonstrates that simply training sentence encoders with contrastive objective, as in previous sentence embedding models, is not sufficient to create an interpretable sentence embedding space. Jointly optimizing both contrastive and generative objectives encourages sentence encoders to preserve sufficient token-level information to better support sentence operations and reconstruction.

## 3.5 Semantic Textual Similarity

**Setup.** In addition to interpretability, we also investigate if INTERSENT preserves the ability to capture semantic similarity. Following previous work, we evaluate our model on the semantic textual similarity (STS) tasks, including STS 2012-2016 (Agirre et al., 2016), STS Benchmark (Cer et al., 2017), and SICK-Relatedness (Marelli et al., 2014). The goal of these tasks is to estimate the semantic similarity between two sentences by computing the cosine similarity of their sentence embeddings. All models are evaluated under the *zero-shot* setting without training on any STS data. We report Spearman's correlation for all tasks.

**Results.** As shown in Tab. 2, incorporating additional properties that support sentence generation leads to a slight performance decrease on the STS tasks compared to the supervised contrastive baseline. We also observe that the gap between unsupervised and supervised contrastive baselines trained on our data is relatively small, as the weak supervision data we use inherently contain some noise. Nevertheless, INTERSENT's performance on STS is still strong enough to match the unsupervised contrastive baseline trained on the same data.

## 3.6 Sentence Retrieval

**Setup.** One important application of sentence embeddings is sentence retrieval, where the goal is to retrieve the most semantically relevant sentence given the query sentence. We conduct sentence retrieval experiments on the QQP dataset, which is originally designed for paraphrase identification. We follow the data splits used in BEIR (Thakur et al., 2021) and report zero-shot performance on the test set that contains 10,000 queries. We use both Mean Reciprocal Rank@10 (MRR@10) and recall@10 as metrics.

**Results.** As shown in Tab. 3, INTERSENT achieves the best performance on the sentence retrieval task. Notably, INTERSENT outperforms the supervised contrastive baseline trained on the same data, which shows that adding interpretability properties can benefit modeling semantic similarity. Combined with the significant improvement in embedding interpretability and strong STS performance, we demonstrate that INTERSENT learns an interpretable sentence representation space that supports various sentence operations while preserving the ability to capture semantic similarity.

## 4 Analysis

To provide a better understanding of INTERSENT, we investigate how INTERSENT handles longer text, and present an ablation study on the effect of individual loss functions, and choice of pre-trained language models for encoders and decoders. Then, we analyze the operator functions learned by INTERSENT through a detailed case study.

## 4.1 Passage Retrieval

The goal of passage retrieval is to retrieve the most semantically relevant passage given the query sentence, whereof the query and passages are of different granularities. Sentence embedding models generally do not perform well on passage retrieval tasks due to their asymmetric nature. Additionally, passages are generally much longer than the query and contain multiple sentences, making it challenging for a sentence embedding model to capture their semantics in the same way as it does for single sentences (Muennighoff et al., 2022). To investigate how well sentence embedding models handle longer text, we evaluate passage retrieval performance on NaturalQuestions (Kwiatkowski et al., 2019) and MSMARCO (Nguyen et al., 2016) datasets, under the zero-shot setting without training on any passage retrieval data.

As shown in Tab. 4, INTERSENT achieves the best performance on both passage retrieval tasks. We can also see a clear performance gap between INTERSENT and baselines trained on the same data. This demonstrates that modeling compositional semantics between sentences helps the model better capture the semantics of longer text, and preserves necessary information for retrieval.

| Model | NQ | MSMARCO |
|---|---|---|
| SimCSE | 41.58 | 35.55 |
| DiffCSE | 40.52 | 30.64 |
| *Encoders trained on our data* | | |
| Unsupervised Contrastive | 43.10 | 35.13 |
| Supervised Contrastive | 42.90 | 32.56 |
| InterSent | **49.64** | **37.87** |

Table 4: Model performance on zero-shot passage retrieval tasks. We report recall@100 on both NaturalQuestions and MSMARCO datasets.

| Model | Interpret. | STS |
|---|---|---|
| INTERSENT | 64.18 | 77.16 |
| - Generative-only | **65.64** | 64.11 |
| - Contrastive-only | - | **78.29** |

Table 5: Model performance on the interpretability and STS tasks trained with different training objectives. We report the average ROUGE-L score on interpretability tasks and the average Spearman's correlation on STS tasks.

## 4.2 Ablation Study

**Effect of Training Objectives.** We conduct an ablation study on the role of contrastive and generative objectives by comparing the model performance of INTERSENT with generative-only and contrastive-only baselines. Both of these two baselines are optimized using only one of the training objectives. For the contrastive-only baseline, we only report the STS performance since the decoder is not aligned to produce any meaningful output given the sentence embedding.

As shown in Tab. 5, the combination of contrastive and generative objectives is crucial to supporting sentence operations while maintaining the ability to capture semantic similarity. Without a generative objective, it is impossible to examine the content being encoded in the sentence embeddings. On the other hand, the generative-only baseline only improves slightly on generative tasks at the cost of a significant performance drop on STS tasks. INTERSENT achieves a desirable balance between interpretability and semantic similarity.

**Choice of Operators.** We investigate the effect of using simple arithmetics instead of MLPs as operators by simply computing addition and subtraction for sentence fusion and difference respectively. The compression operator remains to be trainable MLPs for both models. As shown in Tab. 6, both models have similar STS performance, but defining

| Operator | Fusion | Difference | STS |
|---|---|---|---|
| Arithmetic | 59.02 | 62.28 | **77.46** |
| MLP | **82.19** | **64.34** | 77.16 |

Table 6: Model performance on sentence fusion, difference and STS trained with different choice of operators. We report ROUGE-L score on interpretability tasks and the average Spearman's correlation on STS tasks.

operators with simple arithmetics leads to a significant decrease in generation performance, especially on sentence fusion. This demonstrates that, while simple arithmetics themselves are easier to understand, they do not accurately capture the nature of sentence operations in the embedding space.

## 4.3 Case Study

To better understand the characteristics of operator functions learned by INTERSENT, and how they interact with each other, we conduct a case study on multi-step sentence operations enabled by INTERSENT. We present a few representative examples in Tab. 7, which covers basic operations of our method (fusion, difference, and compression), as well as compound ones that combine two basic operations. All sentence operations we demonstrate are carried out in the sentence embedding space, and the output sentence is decoded from the final embedding calculated by the operators.

As shown in Tab. 7, INTERSENT can generate coherent output sentences that follow the individual sentence operations we apply to the embeddings. Moreover, we observe that the fusion and difference operators learned by our method indeed represent inverse operations on sentences, as shown by the output of two compound operations: *difference after fusion* and *fusion after difference*. Our model does not directly enforce this property. Instead, it emerges as a result of the joint training on sentence fusion and difference tasks. Operators learned by INTERSENT can also be combined in many other ways, and we demonstrate two examples of compound operations supported by INTERSENT: *multi-sentence fusion* that fuses more than two sentences, and *compression after fusion*, which compresses the combined information from two sentences. As shown in Tab. 7, INTERSENT generates reasonable outputs for these compound operations even though they are not directly optimized during training. This demonstrates the potential of our interpretable sentence embedding space in which we can represent more complex and diverse sentence

| Operation | Sentence |
|---|---|
| Fusion | (A) They wanted to do more than just straight news. |
| | (B) They hired comedians who were talented vocalists. |
| | (A ⊕ B) Wanting to do more than just straight news, they hired comedians who were talented vocalists. |
| Difference after Fusion | ((A ⊕ B) ⊖ A) They wanted to hire talented comedians who were more vocal. |
| Difference | (A) The first edition of the dictionary was printed in 1940, but soon became out of print in 1958. |
| Fusion after Difference | (B) The first edition of the dictionary was printed in 1940. |
| | (A ⊖ B) However, it soon became out of print in 1958. |
| | (B ⊕ (A ⊖ B)) The first edition of the dictionary was printed in 1940, but soon it became out of print in 1958. |
| Compression | (A) Nearly one million people have been left without water in South Africa's northern region because of a disastrous drought, the regional Water Affairs acting director said Wednesday. |
| | ($\overline{\text{A}}$) One million people left without water in drought-hit South Africa. |
| Multi-sentence Fusion | (A) Due to financial difficulties, the airline filed for bankruptcy. |
| | (B) The airline suspended scheduled passenger services. |
| | (C) The airline planned to lease its fleet to other airlines. |
| | ((A ⊕ B) ⊕ C) However, due to financial difficulties, the airline suspended scheduled flights for bankruptcy, and the airline planned to lease its fleet to other airlines. |
| Compression after Fusion | (A) The Commonwealth summit opened in Sri Lanka with heavy security on Friday. |
| | (B) The summit was attended by heads of state or their representatives from 53 member nations. |
| | ($\overline{\text{A} \oplus \text{B}}$) The high security summit was convened by leaders of the Commonwealth. |

Table 7: Examples of sentence operations supported by INTERSENT. Following the notation we defined in §2.1, we use ⊕, ⊖ and − to denote sentence fusion, difference and compression respectively.

operations by combining basic operators.

## 5   Related Work

**Sentence Embedding.** Following the distributional hypothesis of semantics (Harris, 1954), early unsupervised sentence embedding methods (Kiros et al., 2015; Hill et al., 2016; Logeswaran and Lee, 2018) extend the idea of word embedding models (e.g., word2vec (Mikolov et al., 2013)) by predicting surrounding sentences based on the given sentence. Supervised methods (Conneau et al., 2017; Cer et al., 2018; Reimers and Gurevych, 2019) utilize human-annotated data, mostly premise-hypothesis pairs from natural language inference, to improve the quality of sentence embedding further. Recently, contrastive learning has emerged as a widely used learning paradigm for sentence embeddings (Giorgi et al., 2021; Yan et al., 2021; Gao et al., 2021; Chuang et al., 2022). These methods learn a well-structured representation space by explicitly bringing sentences with similar semantics (or augmented versions of the same sentence) closer. Meanwhile, several works have also explored generative modeling of sentence embeddings with denoising or masked language modeling objectives (Giorgi et al., 2021; Wang et al., 2021; Huang et al., 2021; Gao et al., 2021; Chuang et al., 2022; Wu and Zhao, 2022). Unlike contrastive learning, purely generative methods do not directly optimize the similarity between embeddings, and generally do not outperform contrastive methods on semantic similarity tasks.

**Representation Interpretability.** Prior works have studied the interpretability of text representations and their operations from various perspectives. Early works on word embeddings (Mikolov et al., 2013; Pennington et al., 2014; Arora et al., 2016; Ethayarajh et al., 2019) have demonstrated compositional properties of word embedding spaces that allow us to interpret simple arithmetic operations as semantic analogies. Similar properties have been studied in the context of sentence embeddings (Zhu and de Melo, 2020). Previous works have also investigated the compositionality of word, and phrase embeddings from pre-trained language models (Yu and Ettinger, 2020; Hupkes et al., 2020; Dankers et al., 2022; Liu and Neubig, 2022). Another important aspect of interpretability is whether the original information can be recovered from its embedding alone. While this generative property has been used as an auxiliary objective to improve sentence embedding models (Wang et al., 2021; Huang et al., 2021; Gao et al., 2021; Chuang et al., 2022; Wu and Zhao, 2022), the quality of the generated text is rarely in the interest of these methods.

# 6  Conclusion

In this work, we present INTERSENT, an end-to-end framework for learning interpretable sentence embeddings that support sentence operations, including fusion, difference, compression, and reconstruction. INTERSENT combines both contrastive and generative objectives to optimize operator networks together with a bottleneck encoder-decoder model. Experimental results show that INTERSENT significantly improves the interpretability of sentence embeddings on four textual generation tasks. Moreover, we demonstrate that our interpretable sentence embedding space preserves the ability to capture semantic similarity, and even improves performance on retrieval tasks.

## Acknowledgement

We appreciate the reviewers for their insightful comments and suggestions. James Huang and Muhao Chen were supported by the NSF Grants IIS 2105329 and ITE 2333736.

## Limitations

First, as sentences can be transformed and combined in much more diverse and complex ways such as multi-sentence intersection (Brook Weiss et al., 2022), the list of sentence operations we study in this work is not exhaustive. Additional constraints, such as the inverse relationship between fusion and difference, may also be introduced to directly enforce the consistency of operators. Second, all training datasets we use are generated automatically thus, they inevitably contain noise. In this regard, our method shares the same limitations as the broad class of weakly supervised methods where training data are automatically generated.

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

# A  Dataset

All training datasets we use in this work are publicly available except Gigaword which is licensed under LDC. We use them in accordance with their license and intended use.

We use the balanced Wikipedia portion of **Discofuse** dataset, which consists of 4,490,803/45,957/44,589 instances for train/dev/test respectively. **WikiSplit** dataset consists of 989,944/5,000/5,000 for train/dev/test respectively. **Google** dataset consists of 200,000/10,000 for train/test respectively. For **Gigaword**, we filter out headline-sentence pairs with fewer than four overlapping tokens after removing stopwords. The resulting dataset consists of 3,535,011/190,034/178,929 for train/dev/test respectively. **ParaNMT** dataset consists of 5,370,128 paraphrase pairs and we use the entire dataset for training.

# B  Hyperparameter

All experiments are conducted on NVIDIA V100 GPUs. Model training takes roughly 15 hours to complete on an 8-GPU machine. INTERSENT uses RoBERTa-base and BART-base as the encoder and

decoder respectively, which has roughly 200 million parameters in total. During training, we apply a linear learning rate schedule with a linear warmup on the first 5% of the data. All inputs are truncated to a maximum of 64 tokens. We finetune the encoder and decoder with a learning rate of 5e-6 and 1e-4 respectively. All operator networks use ReLU as the activation function. The intermediate dimension size of the compression operator is set to 384. We tune all hyperparameters on the STS-B and sentence generation dev sets.

## C    Choice of Pretrained Language Models

We compare the dev set performance of different combinations of pretrained language models as encoders and decodes for INTERSENT in Tab. 8. We observe that the pair of RoBERTa and BART as the encoder and decoder achieves a good balance between generation and semantic similarity tasks.

| Model | Interpret. | STS-B |
|---|---|---|
| BERT + BERT | 36.18 | 78.75 |
| T5 | **60.02** | 79.37 |
| RoBERTa + BART | 55.18 | **81.22** |

Table 8: Model performance on the dev set of interpretability tasks and STS-B. We report the average ROUGE-1 score on interpretability tasks and the average Spearman's correlation on STS-B.