# OpenReview forum: "Bridging Continuous and Discrete Spaces: Interpretable Sentence Representation Learning via Compositional Operations"
_EMNLP/2023/Conference — EMNLP 2023 Main_

### Official Review · Reviewer_KAiQ · 2023-07-27

**Soundness:** 4

**Excitement:**

4: Strong: This paper deepens the understanding of some phenomenon or lowers the barriers to an existing research direction.

**Paper Topic And Main Contributions:**

The paper describes a method for computing sentence embeddings
that aims to be interpretable and that supports compositional operations.

the operations at stake in the paper are
 - sentence fusion
 - sentence difference
 - sentence compression
 - sentence reconstruction

In practice the authors encode sentence with a transformer encoder (BERT) and take  the [CLS] token embedding
  as sentence embedding. They compose sentence embeddings for 2 sentences by taking both embeddings and passing them through a two layer MLP.

The loss is contrastive like in previous work but they also add a generative objective that is shown to be highly relevant for interpretability and system performance.

The paper carries a large number of experiments on benchmark datasets and rely on strong encoder and decoders : BART,ROBERTA and T5. On interpretability datasets the authors report strong improvements over the baselines and strong performances on semantic textual similarity and on sentence retrieval and passage retrieval.

The discussion of the paper motivates the relevance of using two objectives and of using an MLP for composition rather than explicit arithmetic operators. A qualitative illustration of the system output is also provided









**Questions For The Authors:**

This is not exactly a question, but a more detailed qualitative analysis of successes and failure would be enlightening.
you provide essentially successful examples, analysis of failures would be an interesting add-on to this paper .

**Reasons To Accept:**

- The paper relies on a simple idea
- The idea is well motivated by the experiments
- The results are state of the art and sometimes provide a very strong improvement
- The results are reported on a wide range of datasets


**Reasons To Reject:**

- nothing specific

**Reproducibility:**

4: Could mostly reproduce the results, but there may be some variation because of sample variance or minor variations in their interpretation of the protocol or method.

**Reviewer Confidence:**

3: Pretty sure, but there's a chance I missed something. Although I have a good feel for this area in general, I did not carefully check the paper's details, e.g., the math, experimental design, or novelty.

---

> ### Author Rebuttal · Authors · 2023-08-29
>
> ### Question 1
> We appreciate this suggestion, and will accordingly add more examples in the revision.

---

### Official Review · Reviewer_43Hp · 2023-08-02

**Soundness:** 4

**Excitement:**

5: Transformative: This paper is likely to change its subfield or computational linguistics broadly. It should be considered for a best paper award. This paper changes the current understanding of some phenomenon, shows a widely held practice to be erroneous in someway, enables a promising direction of research for a (broad or narrow) topic, or creates an exciting new technique.

**Paper Topic And Main Contributions:**

This paper proposes a novel sentence embedding InterSent both capturing semantic similarity and sentence operation. When applying fusion, difference, and compression operators to textual sentences, one can directly obtain the resulting sentence by manipulating embeddings of the input sentences and conducting reconstruction. For this purpose, the authors parameterize the sentence operation in embedding space by an MLP. During training, the input and output sentences are first fed into the encoder to get their embedding. Then they align the operated embedding with the output embedding by contrastive learning and use a decoder to reconstruct the output sentence from the operated embedding. Extensive experiments demonstrate InterSent significantly outperforms previous embedding methods on operation interpretability while keeping comparative on semantic similarity tasks.

**Questions For The Authors:**

A.	In L268, the authors claim they conduct reconstruction evaluation using ParaNMT. But I don't find relevant results. Do I miss something?

**Reasons To Accept:**

* This work pushes the sentence embedding further than semantic similarity. It inspires a new direction using sentence embedding to solve generative tasks.
* The advantage of InterSent over previous sentence embedding methods on interpretability for the defined three operations is significant.
* The case study is very useful. The results that fusion and difference are mutually inversed operations are impressive.
* The paper is well-organized and easy to follow.


**Reasons To Reject:**

* The definition of fusion and difference is simple in this paper. In the case study, I find the fusion operator almost just concatenates the two sentences, while the difference just removes sentence B. I do not see its value in practice, since for fusion, one can simply reconstruct the two sentences respectively and concatenate them.
* The evaluation datasets are constructed automatically. Hence, it is unclear how InterSent performs in real natural language scenarios.
* Though interesting, I still have no idea about the application of InterSent and don’t understand its relationship with the tasks listed in L68-L72. It would be much better to further elaborate this point.


**Reproducibility:**

4: Could mostly reproduce the results, but there may be some variation because of sample variance or minor variations in their interpretation of the protocol or method.

**Reviewer Confidence:**

4: Quite sure. I tried to check the important points carefully. It's unlikely, though conceivable, that I missed something that should affect my ratings.

---

> ### Author Rebuttal · Authors · 2023-08-29
>
> ### Applications of InterSent
> The tasks of sentence fusion/difference are more than just simple concatenation/deletion, as they usually involve adding connectives (which requires understanding of inter-sentence relations), manipulating clauses, coreference resolution, paraphrasing, etc. In addition to interpretability, InterSent also has value in practice, especially in scenarios where we need to manipulate sentence embeddings (e.g. retrieving summary, searching information related to multiple input sentences), as the resulting vector from a sentence operation can be computed very efficiently without the need to reconstruct all input sentences.
>
> ### Evaluation Datasets
> While the evaluation datasets are automatically constructed to avoid expensive human annotations,  we would also like to point out that these are standard evaluation datasets used in previous works on sentence operations.
>
> ### Question 1
> We only use the ParaNMT as a noisy training set for sentence reconstruction, and do not directly evaluate sentence reconstruction as a downstream task since no sentence operator is associated with this task. However, the quality of sentence reconstruction is reflected in other sentence operation tasks as shown in Table 1, since all of them require reconstruction from sentence embeddings as the final step.

---

### Official Review · Reviewer_eJVG · 2023-08-04

**Soundness:** 3

**Excitement:**

3: Ambivalent: It has merits (e.g., it reports state-of-the-art results, the idea is nice), but there are key weaknesses (e.g., it describes incremental work), and it can significantly benefit from another round of revision. However, I won't object to accepting it if my co-reviewers champion it.

**Paper Topic And Main Contributions:**

In this paper, the author claims that in addition to sentence similarity, compositional operations such as fusion and difference can also represent sentence semantics. Therefore, the paper proposes InterSent, an end-to-end framework for learning sentence representations that supports these compositional operations in embeddings. Specifically, the paper introduces a contrastive objective and a generative objective to ensure that the model captures the compositional operations while preserving the ability to compare similarity.

Unlike traditional Seq2Seq models that attend the decoding tokens to all encoded tokens, InterSent adopts a bottlenet Model that attends the decoding tokens to a single representation extracted by an encoder. This forces the encoded sentence representation to cover as much information as possible. The contrastive objective then pushes the encoded embedding close to the embedding of the sentence processed by the compositional operations. Additionally, the generative objective ensures that the target embedding can be decoded by the sentence embedding.

The proposed InterSent is evaluated on various tasks. The results demonstrate that InterSent outperforms all the baselines by a significant margin. The paper also includes extensive analysis and ablation studies to further validate the effectiveness of InterSent.

**Questions For The Authors:**

See the weakness part, correct me if I made some mistakes.

**Reasons To Accept:**

1. The writing in this paper is clear and engaging, and easy to follow.

2. The results obtained in this study exhibit great strength, particularly in the context of 4 interpretable-related tasks. The improvements achieved are truly impressive.

3. The inclusion of extensive analysis and ablation studies greatly enhances the reliability of this paper.

**Reasons To Reject:**

1. In the experimental results section, the author evaluates the baseline models by equipping them with the same operator and decoder as the proposed InterSent model, and only optimizing the operator and decoder networks. While the author's motivation is to show that the sentence representation extracted by the InterSent encoder performs better than the baselines trained with different objectives, it may be unfair to the baselines for the following reasons:

- The encoder of InterSent is jointly optimized, while all the baseline models are frozen(Correct me if I am wrong for this part). Since the evaluation is conducted on a generation task, regardless of the objectives equipped to the framework, of course the models that freeze the encoder will perform poorly compared to models that optimize both the encoder and decoder. This aspect weakens the persuasiveness of the results.

- The model structure and training objectives are usually complementary. The specific bottleneck + operator + decoder structure in  InterSent may not necessarily be generalizable or suitable for other baselines.

2. In lines 310-322, it would be beneficial to provide more details about the supervised contrastive and unsupervised contrastive baselines. This section is somewhat confusing, as it is unclear whether the unsupervised contrastive baseline refers to unsp. SimCSE and the supervised contrastive baseline refers to sup. SimCSE, both trained on the proposed compositional dataset.

3. A critical disadvantage of this work is that all the baselines only consider the encoders to achieve their goals, so all the optimizating strategies only focus on their encoders. However, in InterSent, additional decoder, operator network, and operation-related data are required.

**Reproducibility:**

4: Could mostly reproduce the results, but there may be some variation because of sample variance or minor variations in their interpretation of the protocol or method.

**Reviewer Confidence:**

4: Quite sure. I tried to check the important points carefully. It's unlikely, though conceivable, that I missed something that should affect my ratings.

---

> ### Author Rebuttal · Authors · 2023-08-29
>
> ### Freezing Baseline Encoder
> As we discussed in section 3.4, the goal of this setting is to demonstrate to what extent we can improve from existing sentence embeddings on the sentence operation tasks without sacrificing downstream performance on semantic similarity tasks. This objective has not been studied in previous works. We have made our best effort to conduct a fair comparison with baselines by using an identical model architecture (RoBERTa encoder + operator + decoder), and training contrastive baselines on our data. Combined with our proposed operator networks and bottleneck structure, further fine-tuning the baseline encoders on generative objectives would essentially turn these baselines into InterSent itself.
>
> ### Clarifications of Unsupervised/Supervised Contrastive Baselines
> The unsupervised/supervised contrastive baselines are similar to unsupervised/supervised SimCSE, trained on our dataset instead. Note that, for the supervised baseline, we create weak supervision sentence pairs from the fusion and difference datasets by concatenating the two shorter sentences, as opposed to the original supervised SimCSE that uses human-written NLI pairs.
>
> ### Goal of InterSent
> The goal of this work is very different from previous sentence embedding methods. In addition to learning sentence embeddings that reflect semantic similarity via some distance measure, we aim to improve the interpretability of the sentence representation by supporting sentence operations including fusion, difference, compression, and reconstruction. We have shown in Table 1 that these desirable properties do not come with previous methods. Hence, it is necessary to introduce new model components and datasets to accomplish this goal.

---

### Meta-Review · Area_Chair_zfs5 · 2023-09-20

**Recommendation:** 4

**Metareview:**

This paper proposes a sentence embedding technique that considers the operability and compositionality of sentences, computes semantic similarity, and offers easy interpretability. The research direction is intriguing, the writing is clear, and the experiments are extensive. We hope that the camera-ready version will address areas the reviewers found unclear or lacking, ensuring the manuscript is valuable for a wide range of readers.

---

### Decision · Program_Chairs · 2023-10-07

**Decision:**

Accept-Main

**Comment:**

This paper proposes a sentence embedding technique that considers the operability and compositionality of sentences, computes semantic similarity, and offers easy interpretability. The research direction is intriguing, the writing is clear, and the experiments are extensive. We hope that the camera-ready version will address areas the reviewers found unclear or lacking, ensuring the manuscript is valuable for a wide range of readers.